# The Impact of Peer Educators or Community Health Workers on the Progress of the UNAIDS 90-90-90 Targets in Africa: A Systematic Review and Meta-Analysis Protocol

**DOI:** 10.3390/ijerph18083917

**Published:** 2021-04-08

**Authors:** Hafte Kahsay Kebede, Hailay Abrha Gesesew, Lillian Mwanri, Paul Ward

**Affiliations:** 1Clinical Pharmacy, Defense University, Addis Ababa 1041, Ethiopia; maymegeltana@gmail.com; 2Public Health, College of Medicine and Public Health, Flinders University, Adelaide 5042, Australia; lillian.mwanri@flinders.edu.au (L.M.); paul.ward@flinders.edu.au (P.W.); 3Epidemiology, School of Health Sciences, Mekelle University, Mekelle 231, Ethiopia

**Keywords:** peer educators, community health workers, UNAIDS 90-90-90 targets, first 90, HIV diagnosis, second 90, ART linkage, third 90, virological failure, virological suppression

## Abstract

Background: Africa is far behind from achieving the Joint United Nations Program on HIV and AIDS (UNAIDS) 90-90-90 targets. Evidence shows that the participation of HIV patients as peer educators and other community health workers is substantially improving the entire HIV care continuum and subsequently the UNAIDS targets. This review aims to provide the best available evidence on the impact of peer educators and/or community health workers for the three targets in Africa. Methods: We will include cohort and experimental studies published in English between 2003 and 2020. Studies which reported interventions for HIV diagnosis, initiation of ART, or virological suppression will be included for review. Three steps searching will be conducted: (i) initial search across Google Scholar, (ii) full search strategy across five databases: MEDLINE, PubMed, CINAHL, SCOPUS and Web of Science, and (iii) screening titles and abstracts. Data will be extracted using standardized instruments from the Joanna Briggs Institute Meta-Analysis of Statistics Assessment and Review Instrument (JBI-MAStARI) and analyzed through narrative synthesis, and meta-analyses and regression. Heterogeneity among quantitative studies will be assessed using Cochran Q test and Higgins I^2^. Ethics: A formal ethical approval will not be required as primary data will not be collected.

## 1. Introduction

HIV/AIDS (human immunodeficiency virus/acquired immunodeficiency syndrome) has been an important public health disease since its emergence three decades ago [1]. According to the 2017 Global Burden of Diseases (GBD 2017), 1.94 million people had new HIV infections, 36.8 million people were infected, and 0.94 million people died due to the virus globally [2]. Africa is disproportionately affected and contributed the highest to these figures, sharing 63% to the total number of new infections and 76% to the total number of deaths [2]. 

Although the introduction of antiretroviral therapy (ART) contributed to the marked reduction in incidence, prevalence and AIDS-related death [3], negative outcomes related to all targets are paramount [4,5,6,7,8,9,10,11,12,13,14,15]. For example, delayed presentation to HIV care [4,5], lost-to-follow-up during pre-ART and ART [6,7], poor ART adherence [8,9], immunologic [10,11], clinical [12,13], treatment [12,13] and virologic [14,15] failures were the challenges even in the era of ART. 

To address the aforementioned challenges in general and end the AIDS epidemic in particular by 2030, the Joint United Nations Program on HIV and AIDS (UNAIDS) [16] and partners launched three new and ambitious goals in Melbourne, Australia in 2014 [17]. The first 90 (HIV diagnosis) aimed at diagnosing (and identifying) 90% of people living with HIV; the second 90 (ART provision) aimed at providing 90% of those diagnosed ART; and the third 90 (virological suppression), aimed at achieving viral suppression for 90% of patients receiving treatment. Levis and colleagues’ [18] described that these targets would equate with 90% of people living with HIV knowing their status, 81% of all HIV positive people receiving ART, and 73% of all HIV positive people having viral suppression. 

Although nearly all countries from low-, middle- and high-income countries subscribed to the UNAIDS program, the performance reported by most low- and middle-income countries is unsatisfactory [2,19]. For example, the GBD 2017 HIV collaborators forecasted that only 54 countries, most of them from high-income countries, will achieve the second 90 by 2020 [2]. At the end of 2017, Sub-Saharan Africa (SSA) performed an overall target of 45-86-76 [19]. 

There are global efforts [20] occurring, although fragmented, to improve these targets. For example, interventions to improve the first 90 include HIV testing services in voluntary HIV counselling testing (VCT), outpatient department (OPD), antenatal care (ANC), tuberculosis (Tb) clinic, and sexually transmitted infection clinic (STI) [21], home-based HIV testing [22,23], and HIV-self testing [24,25]. Interventions to improve the second 90 include *seek-test-treat-succeed* model [26], point-of-care (POC) CD4 testing [27], short message service (SMS) text messages, multiple interventions, cognitive behavioural therapy and supporter interventions were effective interventions for ART adherence [28]. Interventions to improve the third 90 include home-based ART care [29], ‘directly observed therapy (DOT)’ for ART [30], and use of treatment partner-assisted therapy [31].

Recently, there has been a global need to significantly shift tasks and decentralize services to community health workers [32,33]. Evidence shows that involvement of these community health workers in Africa brought improvements in the entire HIV care continuum [32,33]. This was, for example, supported by qualitative participants comprising patients with HIV, community representatives, HIV care providers and program managers from a study in Southwest Ethiopia [34]. The study [34] revealed that trained HIV patients (peer educators, trained patients with HIV who disclose their HIV status publicly) and community health workers would improve the coverage of HIV counseling and testing, ART linkage and retention, a program termed *teach-test-link-trace* model. In Ethiopia, peer educators obtain formal training on basic HIV counselling to counsel other people: to get tested, to link newly diagnosed people to HIV care, and to retain in care [34]. As such, the involvement of community health workers and/or patients with HIV in the management of HIV care and treatment has become increasingly popular [32,33]. In fact, these programs are not only cost effective but also found as effective as facility-based services [30,35].

The available evidence lacks systematic analysis of the interventions for each target. Furthermore, the cross-cutting impact of peer educators or community health workers on the progress towards all UNAIDS targets was not synthesized. For example, the intervention, *teach-test-link-trace* model [34], mentioned above was believed to improve the diagnosis, treatment and virological suppression targets. To our knowledge, there is no systematic review/meta-analysis that synthesized the impact of peer educator or community health workers in Africa. There is one review (published in 2019) which aimed to identify effective community-based initiatives in achieving the UNAIDS targets [36]. However, the context of the published review was global. The following facts make Africa’s context peculiar: (i) most Africans live in rural settings, (ii) there is poor infrastructural access to visit health facilities, and (iii) most Africans cannot afford to visit nurses or general practitioners. Thus, the objective of the present review will be to synthesize the most up to date evidence on the impact of peer educators and/or community health workers on the progress to the three UNAIDS targets in Africa.

## 2. Materials and Methods

### 2.1. Protocol

The protocol has been registered under International Prospective Register for Systematic Reviews (PROSPERO), with PROSPERO number of CRD42019135135.

### 2.2. Population, Intervention and Outcomes

The review will consider cohort and experimental studies conducted in African countries and reporting on (i) adult people living with HIV/AIDS, (ii) peer educators’ or community health workers’ intervention, and (iii) either of the following outcomes: HIV diagnosis (UNAIDS first 90), ART initiation (UNAIDS second 90), and ART adherence and/or virological suppression (UNAIDS third 90).

### 2.3. Study Design

A systematic review and meta-analysis will be conducted on studies conducted in English language between 2003 and 2020. We use 2003 as a start date as most African countries introduced HIV care in 2003.

### 2.4. Search Strategy

A systematic literature search will be conducted aiming to find studies regarding peer educators and community health workers aimed at achieving the three UNAIDS treatment targets for African countries. Additional search will also include literature such as conference presentations, scientific reports and any relevant material published by UNAIDS.

The search strategy will include three steps. Firstly, an initial limited search of Google Scholar to develop key terms for three pre-defined concepts relating to the research question were employed.
Concept 1 (peer educators, HIV patient experts, community health workers, health extension workers, adherence supporters),Concept 2 (HIV care, HIV care continuum, antiretroviral therapy care, UNAIDS 90-90-90, HIV care and treatment cascade, first 90, HIV diagnosis, HIV testing, HIV care presentation, second 90, ART initiation, ART linkage, linkage to care, ART provision, retention in care, attrition, lost-to-follow-up, discontinuation, adherence to Adherence, ART compliance, third 90, virological suppression, viral suppression, virological failure, viral failure, immunological success, immunological failure, clinical success, clinical failure), andConcept 3 (Africa, Sub-Saharan Africa, and ‘list of all African countries’).

Secondly, a full search strategy (Appendix A) will be carried out using all identified keywords and index terms across the five databases: MEDLINE, PubMed, CINAHL, SCOPUS and Web of Sciences. The three concepts, concepts 1, 2 and 3, will be connected by ‘AND’ to run the full search strategy in the five databases. 

Thirdly, screening of titles and abstracts from each database will be performed in order to select relevant titles/abstracts for a full text appraisal. In addition, bibliography of relevant documents will be checked. Hand searches of unpublished studies and different sources of grey literature from ProQuest Dissertations and theses (PQDT), Med Nar, World Bank, World Health Organization (WHO) and UNAIDS report will also be conducted. Finally, the number of relevant documents for review will be schematically presented using the Preferred Reporting Items for Systematic Reviews and Meta-Analyses (PRISMA) guidelines.

### 2.5. Study Selection and Quality Assessment

The authors will screen the search outputs using titles and abstracts and independently assess the full text of all potentially eligible studies to analyze whether they meet the eligibility criteria. Two independent reviewers, HKK and HAG, will assess the selected papers for methodological validity using a standardized Joanna Briggs Institute (JBI) appraisal instruments [37] (Appendix A). The instrument has nine (9) questions for cohort studies, 10 questions for experimental studies and 11 questions for systematic review studies. For each article, we will calculate ratio of methodological quality scores using the number of ‘Y’s (yes) as a numerator, and the sum of ‘U’s (unclear), ‘N’s (no) and ‘Y’s as a denominator. We will exclude ‘NA’ (not applicable) from the ratio calculation. Any discrepancies that arise in the decision to include these studies between the reviewers will be resolved through discussion and consensus. To ensure the quality, the risk of bias—*high*, *moderate*, *low and unclear risk*—will be assessed using the Agency for Healthcare Research and Quality (AHRQ) criteria for experimental studies [38]. The type of biases evaluated in the tool are selection, performance, detection, attrition and reporting biases [39]. *High risk of bias* is implied if there is significant bias, i.e., error in study design, data analysis and reporting that invalidates the findings. *Moderate risk of bias* is demonstrated when there is no enough evidence to invalidate the study but susceptibility of a bias. *Low risk of bias* is indicated if the bias is low and results are valid. *Unclear risk of bias* is assumed if it is difficult to judge the studies as a result of poor report. Authors of primary studies will be contacted via email if there will be a missing or unclear data. We will use the Cochrane tool to assess risk of bias in cohort studies (Appendix A). 

### 2.6. Data Extraction

For papers selected to be included in the review, data will be extracted using the standardized data extraction tool from JBI-MAStARI (Appendix A). The data extracted will include specific details about the authors, populations, sample size, year, country, summary of interventions and outcome.

### 2.7. Types of Interventions and Comparators

The review will consider the impact of peer educators and/or community health workers to improve the first, second and third 90 s of the UNAIDS targets, as compared to routine facility-based services. 

### 2.8. Types of Outcome Measures

The outcomes of the review will be three-fold: (i) uptake of HIV diagnosis and number of people who knew their HIV status (first 90), (ii) number of people who linked to or initiated ART (second 90), and (iii) number of people who were virally suppressed (third 90) or improved their ART adherence. The level of adherence to ART [40] is: (i) ‘good’ if the percentage of prescribed ART intake is 95% or above, and less than three doses out of 30, or less than four out of 60 doses are missed; (ii) ‘fair’ if the percentage of prescribed ART intake is 85% to 95%, and 3–5 doses out of 30 or 4–9 out of 60 doses are missed; and (iii) ‘poor’ if the percentage of prescribed ART intake is below 85%, and six doses and above out of 30 or nine doses and above out of 60 doses are missed. We will use WHO criteria [41] to measure virological suppression, in addition to the national definitions.

### 2.9. Types of Studies

Quantitative studies of good quality published since 2003 will be considered for inclusion. These only include prospective and retrospective cohort studies, randomized controlled trials, non-randomized controlled trials, quasi-experimental studies, and systematic review of cohort and/or experimental studies. We include cohort and experimental studies as they are the most suitable for causal effects of interventions.

### 2.10. Data Management, Analysis and Synthesis

Descriptive and inferential statistical analyses will be used to summarize the data. In the final review, a narrative synthesis of outcomes [37] will be demonstrated along with the interventions of selected studies thematically. We will include the following information to summarize the main data from the included studies: author, study setting, study design, study population, sample size, outcome, intervention and summary of findings.

If available, meta-regression and meta-analyses will be applied to see the association between interventions and UNAIDS targets. Two or more studies will be considered for pooling data for meta-regression and meta-analyses once clinical and statistical heterogeneity are assessed [42,43]. Studies with no, low or medium heterogeneity will be included for the meta-regression and meta-analyses. Clinical heterogeneity will be assessed by the research team. For example, with regard to measurements of inclusion of studies with different measurements of an outcome for meta-analysis (e.g., virological suppression if viral count <1000 cells/mm^3^, <50 mm^3^, or other), we will consider the broader perspective [44]—we will include the number of study participants who diagnosed as virological suppression despite the differences in cutoffs (e.g., the number of patients diagnosed as virological suppression whether the viral count is below 1000 mm^3^ or 50 mm^3^). 

Cochran Q test (chi-square test for heterogeneity or chi-square tests for homogeneity) [43] will be used to assess if the intervention effects of individual studies are farther away from the common effect. The presence of statistical heterogeneity is declared if the alpha level of Cochran Q test is below 10% (*p*-value < 0.1). Higgins I^2^ (I^2^) will be used to quantify the level of heterogeneity. I^2^ value of 0%, 25–50%, and 50–75% are considered as no, low and medium heterogeneity [43]. 

Random effect model will be used where moderate heterogeneity is detected but fixed effect model will be used if no statistical heterogeneity is detected [45,46]. However, we will only consider fixed effect model irrespective of the level of heterogeneity if the number of studies that reported the intervention and UNAIDS target will be small (*n* < 5) [45,46]. Effect sizes expressed as relative risk, hazard ratio and mean difference (for continuous data) (and their 95% confidence intervals) for the experimental and cohort studies [47]. The meta-analysis will be conducted separately for each target and intervention of interest using RevMan-5 Software [48]. A sensitivity test will also be applied by omitting and entering small studies and deviant results from the rest of the studies (outliers). A funnel plot will be used to assess publication bias.

GRADE will be used to assess the strength of the body of evidence, as described in detail elsewhere [49]. After completion of the analysis the review will be reported using the preferred Reporting Items for Systematic Reviews and Meta-Analyses (PRISMA) statement (Appendix A) [50]. 

No patient or public will be involved in the review. We will not require a formal ethical approval for this study as it will not involve collection of primary data. We will use the following medias to disseminate findings of the review: publishing in peer-reviewed journals, presenting at workshops and conference, and sharing findings through a media release.

## 3. Conclusions

This protocol for systematic review of the literature will synthesize the most up to date evidence on the impact of peer educators and/or community health workers on the progress to the three UNAIDS 90 s targets in SSA. This will provide future policy directions to achieve the UNAIDs 90-9-90 targets.

## Data Availability

Data supporting the protocol is reported here and no further data is required.

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
