# Peer review of "The Impact of Peer Educators or Community Health Workers on the Progress of the UNAIDS 90-90-90 Targets in Africa: A Systematic Review and Meta-Analysis Protocol"

_ijerph, 2021, doi:10.3390/ijerph18083917_

Round 1

Reviewer 1 Report

I have a few further comments and/or suggestions for the authors

- Please clarify what types of studies will be eligible for this SR.

- Risk of bias:  The risk of bias tool selected is only relevant for randomized controlled studies. How will the bias in observational studies evaluated.

- Search strategy: What is the “list of all African countries”. Did the authors actually include the list, or did they include this as a term? Please clarify and if a list actually has been used, please include it in the supplementary data/table

Author Response

We have attached the word-by-word response to the reviewer comments. 

Reviewer 2 Report

This is a great piece. Authors have address all comments very well. I would love to see this in print. 

Author Response

Many thanks for your support. 

This manuscript is a resubmission of an earlier submission. The following is a list of the peer review reports and author responses from that submission.

Round 1

Reviewer 1 Report

Thank you for inviting me to peer review the systematic review and meta-analysis protocol entitled "The impact of peer educators or community health workers on the progress of the UNAIDS 90-90-90 targets in Africa: A systematic review and meta-analysis protocol", by Kebede and colleagues.

The aim of a published systematic review & meta-analysis protocol is to prospectively describe the methodology that the investigators intend to use in detail, in order to minimize bias in the conduct of the meta-analysis. However, this manuscript describes the methodology to be used in a very high-level, omitting pertinent information. Therefore, this manuscript is incomplete.

Main Major Comments and/or concerns:

- There are significant differences between the manuscript and the protocol registered with the PROSPERO register. For example, the interventions described in the manuscript (peer educators or community health workers) are not mentioned in the PROSPERO registration. The methodologies described in the two protocols should be identical. The manuscript only allows for a more detailed description of the methods.

- The methodology for searching the literature is not clear. Authors state that they plan to explore relevant keywords in Google Scholar that will be used to build a search strategy, but then state that a complete search strategy is available in the online supplement. Unfortunately, there is no online supplement available to me and I cannot see this strategy, but clarity is definitely lacking here. I would suggest the investigators to develop their final search strategy, to include it in the protocol.

- Most importantly, the outcomes of the systematic review and the meta-analysis are not defined.

Overall, this manuscript has very important limitations and is incomplete

Reviewer 2 Report

Abstract

Lines 14/15 - HIV patients as peer educators (no brackets)

Intro

Maybe in the intro a short description/some info on what a community health and peer educators role is with the HIV patients?

Line 35/36 should read as .... has been an important public health....

Line 38/39 should read as ... 36.8 million people were infected (remove had HIV).

PLEASE clarify lines 49/50 OR rewrite more clearly.

Lines 75/76 must be written clearly without brackets and remove or explain a health "extension" worker. 

Material and Methods

Lines 109/110 needs clarity

Line 111 remove "gray"

Lines 113 to 122 - must be broken down and summarised - as in the current presentation it is "long winded" and creates confusion when reading. 

Line 124 - remove ... The three concepts - Start with ... Concept 1,2

Types of studies

Line 173 - why after 2003 - this review is from 2003?

Lines 187 to 198 should be written more clearly as there is too much of information all "cluttered" together. I suggest take each point and summarize and add full stops so that the information is clear.

Conclusion

Line 214 .....presenting at w/shops (remove on)

Line 215 sharing findings ....

Line 220 - Must be removed and put under Section 2.10

Reviewer 3 Report

I had a great delight to go through this protocol manuscript and to update myself with the recent HIV literature and interventions. This is an excellent write-up. I have no any major comments. I attach the report (manuscript) highlighting a few typos only. Authors have informed readers with relevant details of the research question in the background section, well found method section including the details about how they are going to work out the review with the three strategies mentioned. They have well presented their plan to assess the quality of the included studies. The outcomes they are going to assess and the analysis as mentioned are perfectly established. This is an excellent protocol manuscript worthy to go out for publication. 

Round 2

Reviewer 1 Report

Thank you for submitting the revised ms